# Evaluation of a Murine Model for Testing Antimicrobial Implant Materials in the Blood Circulation System

**DOI:** 10.3390/biomedicines9101464

**Published:** 2021-10-13

**Authors:** Caroline Moerke, Marlen Kloss, Katharina Wulf, Sabine Illner, Sabine Kischkel, Martina Sombetzki, Niels Grabow, Emil Reisinger, Alper Öner, Hüseyin Ince

**Affiliations:** 1Department of Cardiology, University Medical Center Rostock, 18057 Rostock, Germany; caroline.moerke@med.uni-rostock.de (C.M.); alper.oener@med.uni-rostock.de (A.Ö.); 2Division of Tropical Medicine and Infectious Diseases, University Medical Center Rostock, 18057 Rostock, Germany; marlen.kloss@med.uni-rostock.de (M.K.); martina.sombetzki@uni-rostock.de (M.S.); emil.reisinger@med.uni-rostock.de (E.R.); 3Institute for Biomedical Engineering, University Medical Center Rostock, 18119 Rostock, Germany; katharina.wulf@uni-rostock.de (K.W.); sabine.illner@uni-rostock.de (S.I.); sabine.kischkel@uni-rostock.de (S.K.); niels.grabow@uni-rostock.de (N.G.)

**Keywords:** antimicrobial, cardiovascular biomaterial, bloodstream infection, in vivo, infective endocarditis, catheter-related infections

## Abstract

Medical device-related infections are becoming a steadily increasing challenge for the health care system regarding the difficulties in the clinical treatment. In particular, cardiovascular implant infections, catheter-related infections, as well as infective endocarditis are associated with high morbidity and mortality risks for the patients. Antimicrobial materials may help to prevent medical device-associated infections and supplement the currently available therapies. In this study, we present an easy-to-handle and simplified in vivo model to test antimicrobial materials in the bloodstream of mice. The model system is composed of the implantation of a bacteria-laden micro-stent scaffold into the murine tail vein. Our model enables the simulation of catheter-related infections as well as the development of infective endocarditis specific pathologies in combination with material testing. Furthermore, this in vivo model can cover two phases of the biofilm formation, including both the local tissue response to the bacterial biofilm and the systemic inflammatory response against circulating bacteria in the bloodstream that detached from a mature biofilm.

## 1. Introduction

Worldwide, more than 1.7 million cardiovascular devices were implanted each year, and the incidence is expected to rise in the future [1,2]. The foreign materials can trigger a local immune response in the host involving the coagulation cascade, complement system, platelets, as well as immune cells, and provide fertile conditions for bacterial biofilm formation and consequently result in medical device-associated infection [2]. After the initial colonization, adhered bacteria can form a biofilm leading to a contamination of the implant material. This contamination can compromise the functionality and performance of the medical device itself but also affects the integration into the surrounding biosystem. Furthermore, this situation can cause an activation of the inflammatory response and poses a serious risk for the patient in terms of systemic infection, sepsis, and ultimately death [3]. Up to 50–70% of healthcare-associated infections are caused by medical device-associated infection, posing a serious issue for clinical therapies [2]. Bacterial contamination of cardiovascular implants or infections originating from medical device implantation often lead to infective endocarditis (IE), which represents a clinically challenging entity requiring multidisciplinary approaches for the diagnosis and therapy. IE with mortality rates of 20 to 25% remain a clinical problem despite advances in antimicrobial as well as surgical therapies. Implant materials are especially prone to bacterial adhesion and need up to 10,000 times less bacterial exposure to get contaminated in comparison to native materials. A possible reason could be that the artificial implant materials, rather than the native tissue, have no adaptive response towards the bacterial adhesion [4,5,6]. In particular, heart valve leaflets were affected by bacterial colonialization and manifestation of IE. Once a mature bacterial biofilm has established, the bacteria in the biofilm are more resistant to antibiotics and antibiotic treatment is often not efficacious. The last option to eradicate the infection is implant removal because there is the risk of leaving a small fraction of non-growing persister bacterial cells that survive antibiotic therapy and may potentially reconstitute the biofilm after the antibiotic treatment is discontinued [3]. About 10 to 30% of IE cases are caused by Streptococcus and Enterococcus species, whereas *Staphylococcus aureus* (*S. aureus*) is the most frequent pathogen, occurring in 70% of the IE cases, and accounts for around 30% of healthcare-associated infections [2]. Therefore, new implant material strategies or functionalizations are needed to prevent medical device-associated infections and supplement the currently available therapies, especially for cardiovascular implant as well as intravascular catheter-related infections, which is the most common cause of nosocomial bacteremia [7].

To date, only 4 antimicrobial technologies are reported with clinical outcomes for orthopedic implants: silver and iodine coatings, gentamicin poly-D, L-lactide (PDLLA) coating and a fast-resorbable hydrogel coating with Defensive Antibacterial Coating (DAC) based on hyaluronan and poly-L-lactide (PLLA) [2]. In vitro models are not able to simulate the complex biological response to a foreign material in the bloodstream and in particular the elimination of the bacteria by the immune cells. In addition, animal models are commonly used to analyze antimicrobial agents in bacterial endocarditis on native heart valves. These in vivo models are very laborious and did not examine the antimicrobial response of implant materials or test materials in the bloodstream [8,9].

In this study, we describe an in vivo mouse model to test new implant materials in the bloodstream regarding their antimicrobial properties in the bloodstream. This model is easy to handle and involves no unnecessary suffering for the animals because the tested material is implanted minimally invasive into the tail vein as a micro-stent. The setup was created for a basic simulation of an IE situation originating from an infected implant or an intravascular catheter-related infection. Therefore, two clinically critical situations can be modeled in a simplified form: (i) the subsequent situation in IE after a bacterial biofilm has been established and the therapy options get complicated as well as (ii) catheter-related bloodstream infections associated with high mortality and also causing severe comorbidities such as IE.

## 2. Materials and Methods

### 2.1. Implant Design and Production

The used implants were designed as micro-stents in the form of tubes composed of the thermoplastic silicone polycarbonate elastomer (TSPCU; ChronoSil80A, AdvanSource Biomaterials, Wilmington, NC, USA) with a length of 10 mm, an inner diameter of 0.2 and outer diameter of 0.45 mm. The TSPCU micro-stents were generated via extrusion (HAAKE MiniLab II, Thermo Fisher Scientific, Karlsruhe, Germany) and cut with medical scissors (Braun, Melsungen, Germany) to the final length of 10 mm. Afterwards, the micro-stent was spray-coated with poly-L-lactide (PLLA; Resomer L210, Evonik Industries AG, Darmstadt, Germany) dissolved in chloroform (J.T. Baker, Radnor, PA, USA) and dried for 1 week at 37 °C in a vacuum drying chamber. The final PLLA coat thickness was about 20 µm. The diameter of the used micro-stent before and after the coating process was measured with a biaxial laser scanner (ODAC 32 XY, Zumbach Electronic AG, Orpund, Switzerland). The micro-stents were threaded on a 6-0 polypropylene suture material (Prolene, Ethicon, Norderstedt, Germany) for better handling. Micro-stents were sterilized for 10 s in 70% ethanol and rinsed 3 times in 1 mL sterile phosphate buffered saline (PBS, pH 7.4; Thermo Fisher, Schwerte, Germany). Handling of the micro-stents was performed in an aseptic environment.

### 2.2. Animals and Bacterial Culture

All animal experiments were performed according to the German animal protection guidelines that have been approved by the local animal care and use committee (7221.3-1-069/19). Female C57BL/6 mice with an age of 8–12 weeks were purchased from Janvier Labs (Saint Berthevin, France) and used in the experiments. All mice were kept on a standard diet and a 12 h day-night rhythm. A clinical endocarditis isolate of *Staphylococcus aureus* FR20 (gift from Institute for Microbiology University Greifswald) was used as bacterial strain. *S. aureus* was cultured in Luria broth (LB; Sigma-Aldrich, Taufkirchen, Germany) at 37 °C with 150 rpm shaking.

### 2.3. Infection of the Implants

To ensure an exponentially growing bacterial culture, an overnight culture of *S. aureus* was diluted 1:50 in LB. The pre-culture was incubated at 37 °C with 150 rpm shaking until an optical density OD_600nm_ of 0.5–0.7 was reached. Optical density was measured in a 96-well plate (Greiner Bio-One, Frickenhausen, Germany) using a micro-plate reader (FLUOstar Omega, BMG LABTECH GmbH, Ortenberg, Germany). For the assessment of the implant contamination concentration different protocols with varying incubation time and bacteria concentration of the incubation solution were tested. After the infection with the bacteria, the implants were washed 3 times with 1 mL PBS, transferred to a new tube containing PBS with 0.1% Tween-20 (Sigma-Aldrich, Taufkirchen, Germany), placed in a sonification bath (USC300T, VWR International, Darmstadt, Germany) for 5 min and the detached bacteria were determined by plating a serial dilution of the bacterial culture on LB agar plates. For the implant infection, the pre-culture was diluted to an OD_600nm_ of 0.1 or 0.5. The sterilized implants were incubated in 1 mL diluted pre-culture for 30 min at 37 °C and 150 rpm shaking, followed by a wash step of 1 mL PBS repeated 3 times. The incubation of the implants with the diluted pre-culture having an OD_600nm_ of 0.1 resulted in a total bacteria load on the implant of 1 × 10^6^ and with an OD_600nm_ of 0.5 resulting in 5 × 10^6^ CFU (colony forming units). CFUs were determined by plating a serial dilution of the bacterial culture on LB agar plates. Up to the implantation, the infected micro-stents were stored in PBS at 4 °C for a maximum of 3 h.

### 2.4. Implantation Procedure

Female C57BL/6 mice were anesthetized with a mixture of ketamine (75 mg ketamine hydrochloride per kg bodyweight; bela-pharm GmbH, Vechta, Germany) and xylazine (10 mg xylazine hydrochloride per kg bodyweight; Rompun^®^ 2%, Bayer Healthcare GmbH, Leverkusen, Germany) via intra peritoneal application. After reaching a deep unconscious state, the mice were placed in a tail vein restrainer (Tailveiner, AgnThos, Lidingö, Sweden), the tail was disinfected via Octenisept spray (Schülke & Mayr GmbH, Norderstedt, Germany) and the vein was manually compressed before puncture. The implants were injected in the lateral tail vein via a syringe applicator with a 21 G cannula (Braun, Melsungen, Germany). The implants were pushed out by a stainless-steel wire (0.4 mm; Rayher, Laupheim, Germany) fixed at the syringe plunger. The animals were divided into 3 groups: (a) receiving a sterile implant; (b) infected implant with 1 × 10^6^ CFU *S. aureus* and (c) 5 × 10^6^ CFU *S. aureus*. For each group, the implants were kept in the mice for 6 h, 24 h and 48 h with n = 5 animals per group and time. Animals given a sterile implant served as controls.

### 2.5. Bacterial Load Determination

For the analysis of the bacterial load, spleens were homogenized by disruption of the organ with a 5 mL syringe plunger (Braun, Melsungen, Germany) in a 70 µm cell strainer (Falcon, Schwerte, Germany) and flushed with PBS. Serial dilutions of the homogenized organs as well as harvested blood were plated on LB agar plates. The CFU were counted after 24 h incubation at 37 °C.

### 2.6. Cytokine Quantification

The concentration of Interleukin-6 (IL-6), granulocyte colony-stimulating factor (G-CSF) and C-reactive protein (CRP) in the blood plasma was performed via a magnetic bead luminex assay (Bio-techne GmbH, Wiesbaden, Germany). Blood plasma was retained after 10 min centrifugation at 4000× *g* and snap frozen in liquid nitrogen. For the luminex assay, the blood plasma was diluted 1:2 and the assay was performed according to the manufacturer’s instructions. Quantification was calculated using the standards provided and measuring a standard curve according to the manufacturer’s instructions. The readout was done with the Luminex^®^ 100/200™ System and the xPONENT^®^ 3.1.871.0 software (Luminex Corperation, Austin, TX, USA).

### 2.7. Gene Expression Analysis

RNA of the tissue adjacent to the implants was isolated with the NucleoSpin RNA kit (Macherey-Nagel, Düren, Germany). For this propose, the mouse tail was cut into a section retaining only the implant and the tail vein with the implant in these sections was flushed with the lysis buffer at least 3 times using a syringe with a 21 G needle (Braun, Melsungen, Germany). RNA concentration was determined with the Colibri micro-volume spectrometer (Berthold Technologies GmbH & Co.KG, Bad Wildbad, Germany). cDNA synthesis was performed using the RevertAid First Strand cDNA Synthesis Kit (Thermo Fisher Scientific, Schwerte, Germany) following the manufacturer’s instructions. Gene expression analysis was accomplished via quantitative real-time PCR (qPCR) using TaqMan gene expression master mix (Applied Biosystems, Thermo Fisher Scientific, Schwerte, Germany) and TaqMan assays (see Table 1), all purchased from Thermo Fisher Scientific. The qPCR was performed using the QuantStudio 3 (Thermo Fisher Scientific, Schwerte, Germany) under the following reaction conditions: 50 °C for 2 min followed by 95 °C for 10 min, 40 cycles at 95 °C for 15 s, and 60 °C for 1 min. The ΔΔCt method was employed for relative quantification. *Gapdh* served as endogenous control.

### 2.8. Histological Staining

Mouse tail sections were fixed in 4% formaldehyde (Süsse Labortechnik, Gudensberg, Germany) and embedded in paraffin after removal of the bone tissue (USEDECALC solution, Medite, Burgdorf, Germany). Histological sections with a thickness of 4–6 µm were cut with the microtome (Hyrax M55 rotary microtome, Carl Zeiss, Oberkochen, Germany). The slides were deparaffinized by two-time incubation for 10 min in xylene (JT Baker, Phillipsburg, NJ, USA) and rehydrated by a serial dilution of ethanol (100%, 95%, 90%, 80%, 70%) and deionized water for 3 min each. For the Masson Trichrome staining, the Trichrome Stain (Masson) and, for the gram staining, Gram Staining Kit (both Sigma-Aldrich, Taufkirchen, Germany) were used according to the manufacturer’s instructions. The DeadEnd™ Fluorometric TUNEL System (Promega, Walldorf, Germany) enabled labelling of the apoptotic tissue of the samples after following the manufacturer’s instructions. The TUNEL sections were counterstained with DAPI to visualize cell nuclei and embedded in Fluoroshield™ mounting medium (Sigma-Aldrich, Taufkirchen, Germany). Image acquisition was performed with the Primovert inverted microscope equipped with AxioCamMRc (Carl Zeiss, Oberkochen, Germany) for the Masson Trichrome as well as gram staining and with the confocal laser scanning microscope LSM 780 (Carl Zeiss, Oberkochen, Germany) for the TUNEL staining.

### 2.9. Statistical Analysis

For all experiments, the statistical analyses were carried out with GraphPad Prism5 software (GraphPad Software Inc., La Jolla, USA). Results are presented as means  ± SD. Data analyses were performed using the two-tailed Mann–Whitney U test. Asterisks are used in the figures to specify statistical significance (*: *p*  <  0.05; **: *p*  <  0.01; ***: *p*  <  0.001).

## 3. Results

### 3.1. Bacterial Load and Clearance

In this model, controlled contaminated implants were used to determine the inflammatory reaction towards the infected implant. The overall goal is to use this basic model to examine new materials towards their antimicrobial properties. The implants are designed as micro-stents (Figure 1A,B) and for the controlled infection with *S. aureus* two implant contamination concentrations (ICC) were used to model a moderate (1 × 10^6^ CFU, colony forming units) and a high (5 × 10^6^ CFU) contamination or biofilm formation. The infected implants were administrated via a syringe applicator into the tail vein of the mice (Figure 1C). In our experiments the bacteria on the implants were also released and found in the bloodstream after 6 and 24 h but not after 48 h. The bacteria were probably detached by the implantation procedure resulting in the bloodstream circulating bacteria. The circulating bacteria concentrations in the blood of the mice were in accordance with the used ICC (Figure 1D). In contrast to the high ICC, for the low ICC the bacteria in the bloodstream were cleared after 24 h showing only the intended moderate systemic reaction. The spleen as phagocytic filter organ removes bacteria from the bloodstream and displayed high bacterial numbers at 6 h, which reflected the initial loading amount of bacteria. After 24 h, a bacterial load could be detected in the spleens but with an around 2.5 times reduction for the low ICC and a 4 times reduction for the high ICC (Figure 1E).

### 3.2. Inflammatory Cytokines

To analyze the inflammatory reaction towards the infected implant, the blood plasma concentration of the pro-inflammatory cytokines Interleukin-6 (IL-6), C-reactive protein (CRP) and granulocyte colony-stimulating factor (G-CSF) were determined. The blood plasma levels of IL-6 were significantly increased after 6 h with the infected implants with an approximately 4 times rise for the low ICC and a higher rise of around 7 times for the high ICC, showing the highest amount of IL-6. After 24 h, the IL-6 blood plasma concentration decreased up to 3 times for the high ICC and continued to decline until 48 h (Figure 2A). In line with the IL-6 blood plasma concentration, the CRP blood plasma concentrations were increased after 6 h particularly for the high ICC about 2 times compared to uninfected control. After 24 h and 48 h, CRP levels declined with no differences between the uninfected and infected implants (Figure 2B). Blood plasma G-CSF levels started to increase after 6 h and reached their maximum after 24 h (around 9 times increase for low ICC and 11 times increase for high ICC compared to uninfected control) and regressed back to baseline after 48 h (Figure 2C).

### 3.3. Local Endothelium Damage

Despite the probably systemic decline of the immune response, the local endothelial tissue in contact with the infected implant displayed regional inflammation. Figure 3A shows a purulent secretion at the implantation site after 48 h with high ICC. Histological staining with Masson Trichrome revealed that uninfected implants were in close contact with the surrounding endothelium, whereas the infected implants had no direct contact to the surrounding endothelium. In line with the increase of ICC, aggregates were found between the implant and the endothelium (Figure 3B). Gene expression of the VEGF and ICAM-1 in the infected implant contact tissue demonstrated a time as well as ICC-dependent increase (Figure 3C,D). For VEGF, the low ICC showed an 3 times increase after 48 h. High ICC presented around 5 times increase in VEGF gene expression after 24 h and a 7 times increase after 48 h. The gene expression of ICAM was elevated approximately 2 times and 11 times for the low ICC and for the high ICC about 10 times and 22 times after 24 h and 48 h, respectively.

For the visualization of the bacteria at the local implantation site, gram staining was performed. *S. aureus*, as gram-positive bacteria, was colored deep purple for small amounts and black for higher densities. After 6 h, only small appearances are shown in deep purple between the implant and the endothelium. After 24 h, the highest bacteria staining colored in black was found and is shown in Figure 4B. The local tissue damage at the implantation site was visualized by TUNEL (terminal deoxynucleotidyl transferase (TdT) mediated dUTP nick-end labelling) assay, that detects apoptotic cells by the labelling of DNA strand breaks. Figure 4B shows the increase of apoptotic tissue damage with the advanced incubation time as well as the ICC. The higher magnification for the 48 h time point displays a complete distribution of the apoptotic tissue damage around or over the implant-endothelium contact area, especially for the high ICC.

## 4. Discussion

The use of antimicrobial materials could help to reduce implant-related infections and prevent subsequent challenging treatments [10].

The used micro-stent coating material PLLA is a well-known and established coating material for cardiovascular devices and controlled drug release systems [11] including direct mixing or adsorption of antibiotics or drug encapsulation in microspheres or core-shell porous nanofibers to avoid burst-release [12,13,14]. Therefore, this material is an ideal candidate for the validation of our in vivo model for testing antimicrobial materials in the bloodstream for cardiovascular devices. The micro-stent can be coated with other polymeric material or chemically functionalized, e.g., with antibiotics, antimicrobial peptides, or antifouling coatings. Hence, the model is highly adaptable to test different materials in an easy-to-handle and non-laborious manner. This in vivo method enables the imitation of intravascular catheter-related infection and the pathology of an established IE in a simplified form and therefore facilitates investigation of new treatment options as well as material innovations for this clinically difficult condition. Our study covers two crucial and dangerous consequences of the biofilm formation: (i) the local inflammation caused by the adhered bacteria as biofilm and (ii) the systemic inflammation caused by the circulating bacteria reflecting the cell spreading and detachment from the mature biofilm.

The use of a lethal dose of bacteria in the reported range of ~1–3 × 10^7^ CFU [15] was avoided or not desired. Especially considering the higher susceptibility to contamination of implant materials, a formed bacterial biofilm is hard to treat and therefore testing of new implant materials with an already established bacterial biofilm can demonstrate their antimicrobial potential. A lethal dose may obscure the local immune response to the biofilm with the overactive systemic immune response. With the investigated non-lethal ICC, a bacterial clearance of the circulating bloodstream bacteria by the mice was shown after 24 h. IL-6 and CRP as acute phase reactants are elevated in response to infection, with IL-6 as the main inducer of CRP expression [16]. Both cytokines were elevated in response to the infected implants after 6 h. Subsequently, due to the fact that the IL-6 concentration decreased after 24 h, the CRP level also returned to baseline values. G-CSF stimulates the proliferation and maturation of granulocyte precursors and activates granulocyte function, which is essential for the elimination of bacteria during the early state of bacterial infection [17]. This likely reflects the clearance of the bacteria from the bloodstream after 24 h and signals an immune response that is still present but only slightly activated after 48 h. For IE, IL-6 as well as G-CSF levels were reported to increase in the human blood plasma [18]. G-CSF and IL-6 has been shown to be involved in neutrophil production as well. Neutrophils are the immune cells which play an important role in the host defense against microorganisms [19]. Thus, the measured ICC dependent increases of IL-6, CRP as well as G-CSF after 6 h highlight the short systemic activation of the immune response of the mice after receiving the infected implants, which also declined again after clearance of the blood circulating bacteria after 24 h.

The common testing methods for antimicrobial materials are subcutaneous implantations in mice [20,21] or rabbits [22]. However, these models take only into account the local infection with indirect contact to the cardiovascular system. IE is usually simulated in time-consuming rabbit [8], rat [23] or mouse [9] models based on surgical valve trauma by placing a polyurethane catheter at the aortic root and subsequent infection with bacteria. These models do not include the testing of materials.

In addition, we saw in our experiments thrombus aggregates, probably consisting of clotted blood as well as immune cells and bacteria, formed near the infected or injured endothelial site. These clots were also found as vegetations, which is a common pathogenesis of IE. These vegetations can substantially influence the function of the heart valves in IE [10,24] and their constitution as well as formation can also be analyzed with our in vivo model especially regarding the initiation of the local inflammatory response. *S. aureus* is known to induce the cell surface expression of ICAM-1 as well as VEGF secretion in endothelial cells as a pro-inflammatory response to mediate monocyte adhesion and recruitment [25,26]. VEGF is also essential in angiogenesis for promoting endothelial survival [27]. This pro-inflammatory and pro-coagulant phenotype of the endothelial tissue is observed in our in vivo model and is typical for endovascular infections including IE.

Antibiotic-releasing materials are under intensive research but pose the risk of having only a short window of activity and afterwards only deliver subinhibitory doses that can lead to the development of antibiotic-resistant bacteria [28]. The in vivo model presented in this study can be extended to any implantation duration to additionally investigate drug-resistant behavior of the bacteria against the tested material. Antimicrobial material grafting or functionalization implement their antimicrobial function via immobilization of antimicrobial peptides [29] or physicochemical changes such as polycationic [30] or zwitterionic coatings [31] that may interrupt the net negative charge of the bacteria membrane. But under in vivo conditions in the bloodstream, various other physical, chemical and biological conditions may act on these materials that might affect the physicochemical properties and therefore impact the life-span of the antimicrobial properties. These functional analyses can also be performed with the presented in vivo model. Nonetheless, materials preventing the adhesion of bacteria via hydrophobic or nanostructured surfaces [28] cannot be directly tested with the presented in vivo model because it will probably not be possible to sufficiently contaminate these implants with bacteria. But the in vivo conditions affecting the surface modification and its antimicrobial life-time can be simulated by this model and the antimicrobial properties can be examined in vitro after the explantation. Metal-based antimicrobial coatings, e.g., silver or copper coatings can also be tested with this in vivo model. Silver is one of the most investigated antimicrobial agents and is incorporated in materials in elemental, salt, complexed or nanoparticle forms. The silver cations disrupt the function of the bacterial cell membrane and metabolic proteins. But regarding the low threshold concentration for cytotoxicity and the low bio- as well as hemocompatibility index of the metal-ions, this poses some greater difficulties for these coatings in the application of cardiovascular devices [28,29]. Finally, various bacterial strains can be investigated with the presented in vivo model and also compared between each other. Therefore strain-specific antimicrobial differences can be highlighted and analyzed for their mechanisms in causing and developing IE.

After all, our simplified model has limitations regarding the direct translation of the results to humans. The human immune system is more complex than the immune system of mice [32]. In the case of catheter-related infection, scar tissue formed around the injection site may also have an impact on the infection event and the body’s response to the injected material. The relation between the bacteria number that burdens the host also cannot be transferred in its entirety from mice to humans. However, results from a mouse model can provide indications of possible responses in the human organism.

## 5. Conclusions

In the end, the presented in vivo model offers an easy-to-handle and simplified setup for testing antimicrobial materials in the bloodstream during two steps of the bacterial biofilm formation. In addition, this model allows an in vivo simulation of catheter-related infections as well as the local pathologies of IE. To the best of our knowledge, this is the first model that enables material testing in the bloodstream in an IE projecting model.

## Figures and Tables

**Figure 1 biomedicines-09-01464-f001:**
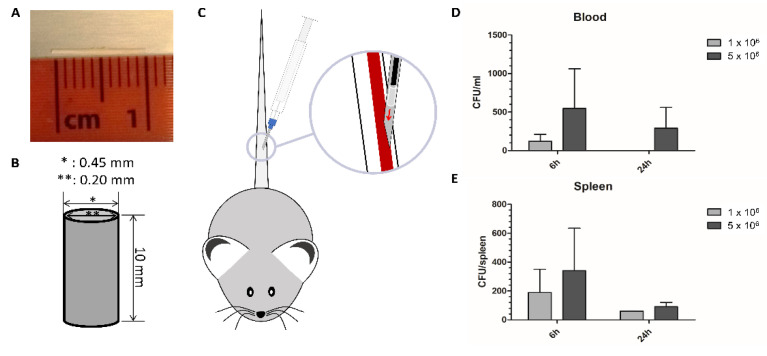
The implant used for the testing of antimicrobial material properties in the bloodstream is (**A**) photographically and (**B**) schematically illustrated (*: uter diameter, **: inner diameter) as well as the implantation procedure (**C**). Bacterial load of (**D**) the blood and (**E**) the spleen are shown by colony forming units (CFU)/mL or per organ for the implants with low (1 × 10^6^) and high (5 × 10^6^) implant contamination concentration.

**Figure 2 biomedicines-09-01464-f002:**
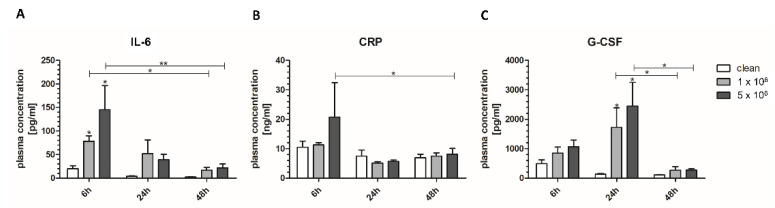
Blood plasma levels of the cytokines (**A**) Interleukin-6 (IL-6), (**B**) C-reactive protein (CRP) and (**C**) granulocyte colony stimulating factor (G-CSF) measured via luminex assay. Statistical analyses above the bars are in reference to the clean control sample at the same time point.*: *p* < 0.05, **: *p* < 0.01.

**Figure 3 biomedicines-09-01464-f003:**
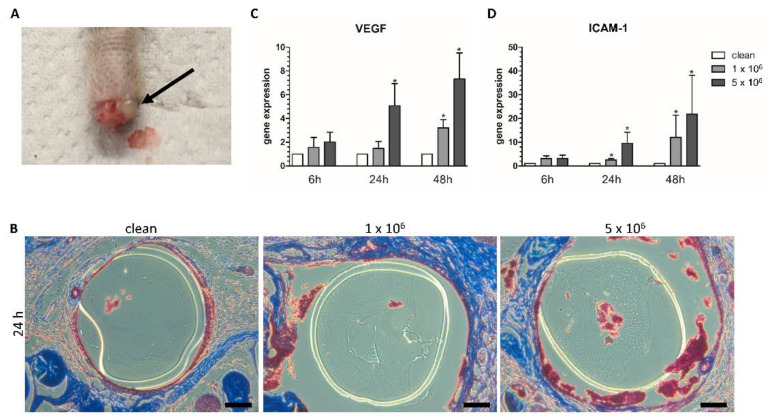
Local endothelium damage originating from the infected implants. (**A**) Purulent secretions at the implantation site after 48 h with 5 × 10^6^ CFU (colony forming units) infected implant. (**B**) Histological staining with Masson Trichrome after 24 h, bars 50 µm. Gene expression of (**C**) vascular endothelial growth factor A (VEGF) and (**D**) Intercellular Adhesion Molecule 1 (ICAM) after 6 h, 24 h and 48 h, *: *p* < 0.05.

**Figure 4 biomedicines-09-01464-f004:**
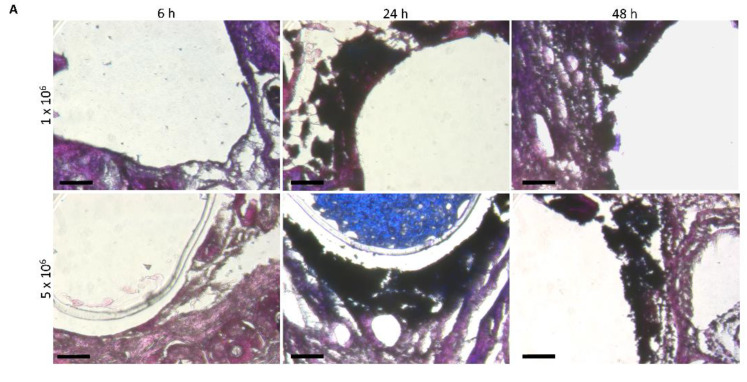
(**A**) Visualization of bacteria at the implantation site after 6 h, 24 h and 48 h. *S. aureus* were stained in black via gram staining. Bars 20 µm. (**B**) Local endothelium apoptotic tissue damage originating from the infected implants. TUNEL staining of the implantation site after 6 h, 24 h and 48 h with 1 × 10^6^ (upper row) as well as 5 × 10^6^ (lower row) CFU (colony forming units) infected implant. Apoptotic cells stained in green and nuclei in blue via DAPI. Bars 100 µm for 10× magnification and 50 µm for 20× magnification.

**Table 1 biomedicines-09-01464-t001:** List of TaqMan assays (Thermo Scientific) used in this study.

Gene Symbol	Gene Name	Assay ID
*Gapdh*	glyceraldehyde 3-phosphate dehydrogenase	Mm99999915_g1
*Icam-1*	intercellular adhesion molecule 1	Mm00516023_m1
*Vegf*	vascular endothelial growth factor A	Mm00437306_m1

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
