# Peer review of "Evaluation of a Murine Model for Testing Antimicrobial Implant Materials in the Blood Circulation System"

_biomedicines, 2021, doi:10.3390/biomedicines9101464_

Round 1

Reviewer 1 Report

  The authors submitted a manuscript investigating the antimicrobial properties of the tubular micro-stents in mice. The authors confirmed the role of micro-stents in terms of bacterial load, inflammatory cytokines, and local endothelium damage through in vivo animal experiments.

  Biofilms are communities of microorganisms that are attached to a surface and play a significant role in the persistence of bacterial infections. Biofilm formation and its associated infections are critical to the success use of implantable medical devices. Studies have shown that different species of bacteria use different "techniques" to approach and attach to surfaces, colonize them, grow and multiply, and eventually form biofilms. There are many factors affecting biofilm formation, and the molecular details of biofilm formation are not fully understood. The contents of the authors' manuscript do not fully simulate or represent the real situation of the biofilm formation of various bacteria in vivo. This so-called model is not as advantageous or able to simulate as many scenarios as the authors claim. The authors are somewhat exaggerating about the significance of the manuscript or the possibility of its future application. The authors should be more objective in evaluating the significance and value of this study.

  Some specific comments are listed below.

  1. In the Results section, some of the descriptions of the results might have been better if they were quantitative.

  2. In the Discussion section, some of the descriptions are a reintroduction of background knowledge and a re-description of the results.

  3. The English needs to be improved to a certain extent. There are more minor errors in grammar and format in the whole manuscript: inconsistencies; tense; spelling mistakes; single and plural expressions; the use of prepositions and definite/indefinite articles; punctuation. For examples:

  anti-microbial, antimicrobial;

  37°C, 37 °C;

  C57BL/6, C57Bl/6;

  3 times, 3-times;

  blood-stream, bloodstream, blood stream;

  **: p < 0.02, **: p < 0.01;

  S.aureus, S.aureus;

  a, b, c…(lowercase letters in the text), A, B, C…( uppercase letters in figures and figure legends);

  ……

  In Page 1, “a clinically challenging entities” should be changed into “a clinically challenging entity”;

  In Page 2, “most frequently pathogen” should be changed into “most frequent pathogen”;

  In Page 4, “Luminex Coorperation” should be changed into “Luminex Corporation”;

  In Page 4, “theses sections” should be changed into “these sections”;

  In Page 7, “After 6 h only” should be changed into “After 6 h, only”;

  In Page 8, “coasts” should be changed into “costs”;

  ……

  The authors should check the full text carefully and correct the mistakes.

  4. The primers used in qPCR should be listed in a table.

Author Response

Dear Reviewer 1,

Thank you for the careful reading of our manuscript and your helpful comments. We have tried to address all of your remarks by changing the manuscript. Please find our corrections below or in the attachment with the edited manuscript .

  1. In the Results section, some of the descriptions of the results might have been better if they were quantitative.

Thank you for the suggestion, we have added some quantitative descriptions to our results to make them more comprehensive.

 - Section 3.1:            “After 24 h, a bacterial load could be detected in the spleens but with an around 2.5 times reduction for both the low ICC and a 4 times reduction for the high ICC (Fig. 1E).
 - Section 3.2:            “The blood plasma levels of IL-6 were significantly increased after 6 h with the infected implants with an approximately 4 times rise for the low ICC and a higher rise of around 7 times for the high ICC, showing the highest amount of IL-6.
In line with the IL-6 blood plasma concentration, also the CRP blood plasma concentrations were increased after 6 h particularly for the high ICC about 2 times compared to uninfected control.”
Blood plasma G-CSF levels started to increase after 6 h and reached their maximum after 24 h (around 9 times increase for low ICC and 11 times increase for high ICC compared to uninfected control) and regressed back to baseline after 48 h (Fig. 2C).”
 - Section 3.3:For VEGF, the low ICC showed an 3 times increase after 48 h. High ICC presented around 5 times increase in VEGF gene expression after 24 h and a 7 times increase after 48 h. The gene expression of ICAM was elevated approximately 2 times and 11 times for the low ICC and for the high ICC about 10 times and 22 times after 24 h and 48 h, respectively.”

  1. In the Discussion section, some of the descriptions are a reintroduction of background knowledge and a re-description of the results.

Thank you for this hint. We have shortened the discussion and removed sentences with general background knowledge. Now, we hope to provide a better summary of our findings in debate to the current state of knowledge. We apologize for the extensive background knowledge explanations but we wanted to facilitate the understanding for readers not so familiar with this topic

We deleted or shortened the following sentences in the discussion:
Bacterial biofilm formation on implants is a huge challenge for the current health care systems especially for cardiovascular implants because of their high complication and morbidity risks during revising operations. [3], [10] Systemic administration of antibiotics to fight implant-related infection is not always successful because bacteria in biofilms are less receptive for antibiotics. Furthermore, long-term antibiotic treatment has numerous side effects such as the destruction of the natural intestinal flora or the induction of antibiotic resistances. Ultimately, the infected device fails in its function and needs to be removed to eradicate the infection. [3]”
In addition, various antibiotics incorporation approaches are reported…”
The advantages beneath the minimal-invasive setup are also the option to study the systemic inflammatory response towards the circulating bacteria in the bloodstream as well as the local endothelial tissue damage.”
“…as the most common cause of nosocomial bacteremia, which is associated with high morbidity and mortality rates and high health care coasts. [7] Moreover, this presented in vivo model also allows simulating…
Biofilm formation passes different phases, after bacteria adhesion and biofilm formation, maturation with bacteria cell spreading and detachment of bacteria from the biofilm. [2]”
IL-6 is involved in the host response to infection and regulates the acute-phase response.”
G-CSF is a major regulator of neutrophil production
In this study, we established an in vivo mouse model to test anti-microbial materials in an easy-to-handle setup.”
“…and has been found to have a chemoattractant effect on immune cells.”
“…leading to cell lysis and death

In addition, we have added some comments regarding the limitations of our study:
After all, our simplified model has limitations regarding the direct translation of the results to humans. The human immune system is more complex than the immune system of mice. [32] In the case of catheter-related infection, scar tissue formed around the injection site may also have an impact on the infection event and the body's response to the injected material. The relation between the bacteria number that burden the host also cannot be transferred in its entirety from mice to humans. However, results from a mouse model can provide indications of possible responses in the human organism.”

Furthermore, we supplemented in the manuscript that our study only shows an IE and catheter-related infection model in a simplified form to avoid exaggerating the significance and value of this study.
 - Abstract:In this study, we present an easy-to-handle and simplified in vivo model…
 - Introduction:The setup was created for a basic simulation of an IE situation originating from an infected implant or an intravascular catheter-related infection.”, “Therefore, two clinically critical situations can be modeled in a simplified form: (i)…
 - Results: “. The overall goal is to use this basic model to examine new materials…
 - Discussion:This in vivo method enables the imitation of intravascular catheter-related infection and the pathology of an established IE in a simplified form,…”, “After all, our simplified model has limitations regarding…“
 - Conclusion:In the end, the presented in vivo model offers an easy-to-handle and simplified setup for testing antimicrobial materials…

  1. The English needs to be improved to a certain extent. There are more minor errors in grammar and format in the whole manuscript: inconsistencies; tense; spelling mistakes; single and plural expressions; the use of prepositions and definite/indefinite articles; punctuation. The authors should check the full text carefully and correct the mistakes.

Thank you very much for the mindful reading and finding of the mistakes. We are sorry for the format inconsistencies, which should not have happened. We corrected all of your findings and had a native speaker checked our manuscript to make sure the English language and style have improved. The corrections are highlighted by the tracking modus.

  1. The primers used in qPCR should be listed in a table.

We added Table 1 in the Material and Method section which lists the used qPCR primer.

We gladly acknowledge your comments and thank you again for the critical review that helped us to improve our manuscript.

Best regards

Caroline Mörke

Reviewer 2 Report

Thank you for providing me the opportunity to review the paper entitled "Evaluation of a murine model for testing anti-microbial implant material in the blood circulation system" by Moerke et al.

General comments:

The study described an in vivo mouse model to test new implant materials regarding their anti-microbial properties in the bloodstream. The authors suggested that two clinically critical situations can be modeled: (i) the subsequent situation in infective endocarditis (IE) after a bacterial biofilm has been established and the therapy options get complicated as well as (ii) catheter-related bloodstream infections associated with high mortality and also causing severe comorbidities such as IE. And concluded that the presented in vivo model offers an easy-to-handle setup for testing antimicrobial materials in the bloodstream during two steps of the bacterial biofilm formation and, this model allows an in vivo simulation of catheter-related infections as well as the local pathologies of IE. It is very interesting and a well-conducted study, and the manuscript itself is well written and the findings are well presented. I have some minor comments.

Minor:

  1. in Figure 4, the authors did gram staining for visualization of bacteria at the local implantation and did TUNEL (terminal deoxynucleotidyl transferase (TdT) mediated dUTP nick-end labelling) assay to detect apoptotic cells. Figure 4 was well presented. I just wonder if the authors did gram staining and TUNEL assay in control group to make sure that the control group implantation site was not infected or contaminated by S. aureus. Because, as you know, it’s very easy to contaminate the implantation site with S. aureus, even though we tried to clean up the op site.
  2. the authors measured bacterial concentrations and all inflammatory markers including IL-6 etc. at 6 hrs, 24hrs and 48hrs only. I just wonder if the authors did measure those at long-term f/u, for example. After 1 week or 15days.
  3. I think there are some limitation in this study. For example, this in-vivo study was using mouse model. It may differ from humans with healed device implantation scar beyond time period. However, there was no limitation comments in this study. Please add on those comments.

Author Response

Dear Reviewer 2,

thank you very much for the review of our manuscript and your helpful comments. We tried to address all your remarks. Please find our answers below or in the attachment with the edited manuscript.

  1. in Figure 4, the authors did gram staining for visualization of bacteria at the local implantation and did TUNEL (terminal deoxynucleotidyl transferase (TdT) mediated dUTP nick-end labelling) assay to detect apoptotic cells. Figure 4 was well presented. I just wonder if the authors did gram staining and TUNEL assay in control group to make sure that the control group implantation site was not infected or contaminated by S. aureus. Because, as you know, it’s very easy to contaminate the implantation site with S. aureus, even though we tried to clean up the op site.

Figure 1 shows the Gram staining of the uninfected implants after 6 h, 24 h and 48 h implantation time. We could not detect any bacteria in the Gram staining of the control samples. During the implantation procedure, we try to take all possible precautions to avoid contamination of our samples, including cleaning the surfaces and the used equipment after each implantation as well as using sterile single-use materials.

Figure 1: Gram staining of the clean, uninfected implants after 6 h, 24 h, 48 h. Bar 20 µm.

Figure 2 exemplary presents the TUNEL staining of uninfected implants after 6 h and 48 h implantation time. We observed no severe tissue damage, visualized by the TUNEL staining, in the control samples as in the infected samples.

Figure 2: TUNEL staining of the clean implants after 6 h and 48 h. Bar 100 µm. Apoptotic cells stained in green and nuclei in blue via DAPI.

  1. the authors measured bacterial concentrations and all inflammatory markers including IL-6 etc. at 6 hrs, 24hrs and 48hrs only. I just wonder if the authors did measure those at long-term f/u, for example. After 1 week or 15days.

Unfortunately, we did not investigate the inflammatory markers later than 48 h. Because of the decay of the inflammatory response after 48 h, we decided not to extent the implantation time. In accordance with the 3R principles (replacement, reduction, refinement) in animal research, we wanted to keep the number of animals at a minimum and included no long-term follow up.

  1. I think there are some limitation in this study. For example, this in-vivo study was using mouse model. It may differ from humans with healed device implantation scar beyond time period. However, there was no limitation comments in this study. Please add on those comments.

Thank you very much for this comment. We have added some remarks regarding the limitations of our model at the end of the discussion.
“After all, our simplified model has limitations regarding the direct translation of the results to humans. The human immune system is more complex than the immune system of mice. [32] In the case of catheter-related infection, scar tissue formed around the injection site may also have an impact on the infection event and the body's response to the injected material. The relation between the bacteria number that burden the host also cannot be transferred in its entirety from mice to humans. However, results from a mouse model can provide indications of possible responses in the human organism.”

We gladly acknowledge your comments and thank you again for the critical review that helped us to improve our manuscript.

Best regards

Caroline Mörke

Round 2

Reviewer 1 Report

  The authors submitted a revised manuscript and made some corresponding changes based on the reviewers' comments. Any change, even a letter or a punctuation mark, should be highlighted for a revised manuscript. However, in the PDF version of this revised manuscript, only a few changes involving references are marked in red, while all other changes are unmarked. Although the authors mentioned “We corrected all of your findings and had a native speaker checked our manuscript to make sure the English language and style have improved. The corrections are highlighted by the tracking modus.” in the response, the revised manuscript still contains some errors.

Author Response

Dear Reviewer 1,

we are very sorry that you did not received the manuscript with the tracking mode. Therefore, we have attached both the first revision of the manuscript with the tracking mode as well as the second revision of the manuscript with the tracking mode as PDF version in the attachment. We checked the manuscript  again and highlighted the changes for you.

We included the following changes in the second revision:

Title:

Evaluation of a murine model for testing antimicrobial implant materials in the blood circulation system

Abstract:

…including both the local tissue response to the bacterial biofilm and the systemic inflammatory response against circulating bacteria in the bloodstream circulating bacteria that detached from a mature biofilm.

Introduction:

Bacterial contamination of cardiovascular implants or infections originating from  medical device…

IE with mortality rates of 20 to 25% remain clinically problematic a clinical problem despite…

Therefore, new implant material strategies or functionalizations are needed to prevent medical device-associated infections and supplement the currently available therapies, especially for cardiovascular implant as well as intravascular catheter-related infections, which is the most common cause of nosocomial bacteremia.

“…the tested material is implanted minimally non-invasively into the tail vein as a micro-stent.

Material and Methods:

Micro-stents were sterilized for 10 sec in 70% ethanol and rinsed 3- times in 1 ml sterile…

“…the tail was disinfected via Octenisept spray (Schülke & Mayr GmbH, Norderstedt, Germany) and the vein was manually compressed before puncture.”

Synthesis Kit (Thermo Fisher Scientific, Schwerte, Germany) following the manufacturer’s instructions
“List of TaqMan assays…”

Results:

(Fig. 1A,B) and for the controlled infection with S. aureus

The circulating bacteria concentrations in the blood of the mice were in accordance to with the used ICC (Fig. 1D).

After 24 h, the IL-6 blood plasma concentration decreased up to 3 times for the high ICC and continued to decline until 48 h.

After 24 h, the highest bacteria staining colored in black was found and is shown in Figure 4B.”

Discussion:

The use of antimicrobial materials could help to reduce implant-related infections and prevent  subsequent challenging treatments. [10]

The used micro-stent coating material PLLA is a well-known and established coating material for cardiovascular devices  and controlled drug release systems...

The micro-stent can be coated with other polymeric material or  chemically functionalized,…

This in vivo method enables the imitation of intravascular catheter-related infection _and the pathology…

IL-6 and CRP as acute phase reactants  are elevated in response to infection…

These models do not include the testing of materials..

IE is usually simulated in a time-consuming rabbit [8], rat [23] or mouse [9] models…

VEGF is also essential in angiogenesis for the promoting of endothelial survival…”

“…especially regarding of the initiation of the local inflammatory response.

Finally, all various bacterial strains can be investigated…

The relation between the bacteria number that burdens the host…

We thank you again for taking the time to carefully review our manuscript and hope to address all your remarks.

Best wishes

Caroline Mörke
